# Preparing GIS data for analysis of stream monitoring data: The R package openSTARS

**Mira Kattwinkel**[1]*, **Eduard Szöcs**[1¤], **Erin Peterson**[2,3], **Ralf B. Schäfer**[1]

**1** Institute for Environmental Sciences (iES), University of Koblenz-Landau, Landau, Germany, **2** Institute for Future Environments, Queensland University of Technology, Brisbane, Australia, **3** Australian Research Council Centre of Excellence in Mathematical and Statistical Frontiers (ACEMS), Brisbane, Australia

¤ Current address: BASF SE, Biostatistics & Data Sciences, Ludwigshafen, Germany
* mira.kattwinkel@gmx.net

## Abstract

Stream monitoring data provides insights into the biological, chemical and physical status of running waters. Additionally, it can be used to identify drivers of chemical or ecological water quality, to inform related management actions, and to forecast future conditions under land use and global change scenarios. Measurements from sites along the same stream may not be statistically independent, and the R package SSN provides a way to describe spatial autocorrelation when modelling relationships between measured variables and potential drivers. However, SSN requires the user to provide the stream network and sampling locations in a certain format. Likewise, other applications require catchment delineation and intersection of different spatial data. We developed the R package openSTARS that provides the functionality to derive stream networks from a digital elevation model, delineate stream catchments and intersect them with land use or other GIS data as potential predictors. Additionally, locations for model predictions can be generated automatically along the stream network. We present an example workflow of all data preparation steps. In a case study using data from water monitoring sites in Southern Germany, the resulting stream network and derived site characteristics matched those constructed using STARS, an ArcGIS custom toolbox. An advantage of openSTARS is that it relies on free and open-source GRASS GIS and R functions, unlike the original STARS toolbox which depends on proprietary ArcGIS. openSTARS also comes without a graphical user interface, to enhance reproducibility and reusability of the workflow, thereby harmonizing and simplifying the data preprocessing prior to statistical modelling. Overall, openSTARS facilitates the use of spatial regression and other applications on stream networks and contributes to reproducible science with applications in hydrology, environmental sciences and ecology.

## Introduction

Streams and rivers are regularly monitored to assess their biological (e.g. species composition or abundance), chemical (e.g. nutrient or pesticide concentrations) and physical (e.g. temperature) status. For example, the EU Water Framework Directive's (WFD) aim to maintain and

**Data Availability Statement:** The R package openSTARS can be downloaded from the Comprehensive R Archive Network (CRAN) at https://cran.r-project.org/web/packages/openSTARS/index.html and also at github https://

github.com/MiKatt/openSTARS. All data used in
the example workflow are available within the
package.

**Funding:** MK was partly funded by the EU-
INTERREG V Upper Rhine via project 1.6 SERIOR
(Security- Risk-Orientation). The funders had no
role in study design, data collection and analysis,
decision to publish, or preparation of the
manuscript. BASF SE provided support in the form
of salaries for author ES at time of preparation of
the manuscript, but did not have any additional role
in the study design, data collection and analysis,
decision to publish, or preparation of the
manuscript. The specific roles of these authors are
articulated in the 'author contributions' section. All
other authors received no specific funding for this
work. There was no additional external funding
received for this study.

**Competing interests:** ES commercial affiliation
does not cause any competing interests and it also
does not alter our adherence to PLOS ONE policies
on sharing data and materials.

improve water quality led to vast monitoring efforts to assess the status of European water bodies, comprising a monitoring network of more than 67000 sites in 2012 [1]. This extensive network is complemented by additional national, regional or local stream monitoring programs; for example to evaluate pesticide concentrations or to monitor industrial discharge [2]. Monitoring data is often related to climatic, land use or hydrological predictors to investigate the effects anthropogenic impacts on in-stream condition or to support biodiversity conservation [3, 4].

Sampling sites in branching stream networks are often connected by stream flow and may share similar landscape characteristics (e.g. elevation or climate) to sites in close geographic space. Therefore, the measurements may be correlated in geographic space, topological space, or both [5]. This violates the assumption of independence in many classical statistical approaches (e.g. linear regression) and alternative methods accounting for the spatial dependence in the data should be used [6]. The package SSN [7] for R statistical software [8] provides the functionality to fit spatial statistical stream network (SSN) models using a mixture of covariance functions that account for the unique spatial relationships found in streams data [9]. However, several data preparation steps are necessary to generate the spatial information needed to fit these models.

SSN models have been applied in almost 50 case studies mainly predicting water temperature [10] but also other physico-chemical [11] or biological variables [12, 13]. To the best of our knowledge, all of the applications used the Spatial Tools for the Analysis of River Systems (STARS) toolbox [14] for preparing the spatial input data to allow for subsequent modelling in SSN. Although STARS is freely available, it depends on the proprietary software ArcGIS [15], which does not allow users to study or improve the source code and incurs relatively high license costs [16]. Additionally, a redesign of the ArcGIS Pro environment and discontinuation of the personal geodatabase would require significant modification of STARS to meet the requirements of the latest ArcGIS versions.

Here, we introduce the R package openSTARS [17] as an alternative tool for data preparation of spatial stream network data, which can subsequently be used with the SSN package and other applications. The package is independent of proprietary software, relying on the geographic information system (GIS) functionalities of R and GRASS GIS via the package rgrass7 [18]. GRASS GIS is free and open source software (FOSS) with a strong user and developer community and offers powerful functions for deriving stream networks and catchment delineation [19]. Our implementation within R also releases the user from the need to familiarize themselves with GRASS GIS. We provide example code in the S2 File, enabling readers to recreate this workflow using their own stream data, and compare openSTARS with STARS output.

## Material and methods

### Background

The openSTARS package provides functions to generate the spatial information needed to fit SSN models to stream data using the SSN package [7] (Fig 1). The STARS toolbox uses and cleans an existing stream network in vector format, whereas openSTARS creates the stream network from a digital elevation model (DEM) based on the GRASS functions r.watershed and r.stream.extract [20]. Optionally, an existing stream network can be provided in vector format that guides the stream network derived from the DEM. In the SSN package, several topological conditions are inadmissible [14]: converging nodes (two stream segments converge at a confluence without flowing into another downstream segment), diverging nodes (a stream segment flows into a node and splits into multiple segments downstream of the node), and

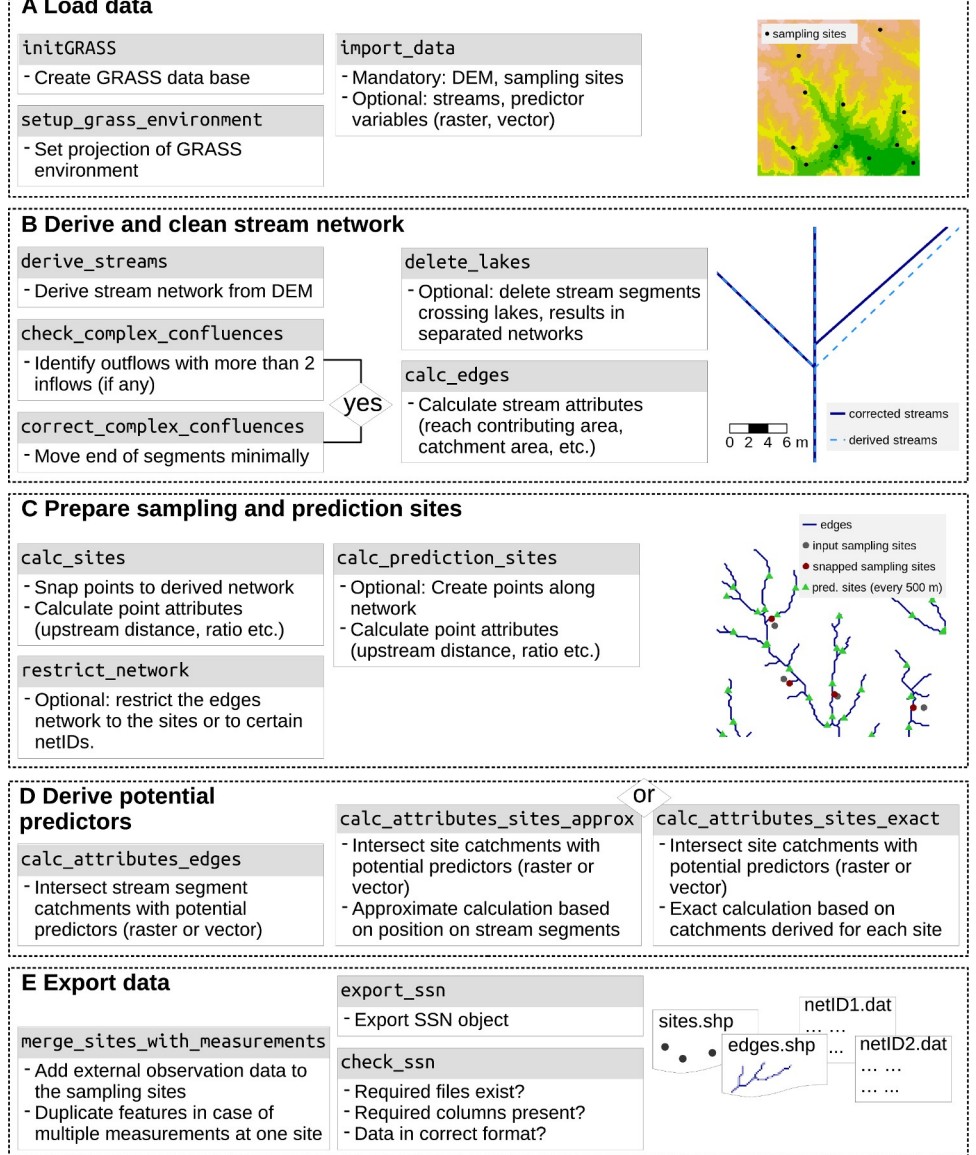

**Fig 1. openSTARS workflow.**

complex confluences (more than two stream segments flow into a node, and out to a single downstream segment). A major advantage of relying on a streams dataset derived from a DEM is that it is free of true topological errors, i.e. all streams flow downstream, there are no dupli-cate reaches and only a single outflow per network. The algorithms also produce non-braided networks, which would lead to diverging nodes. Hence, this approach can save a significant amount of editing time when there are numerous topological errors in a vector stream network.

## Workflow

**Load data.**   First, the GRASS environment is set based on the region and projection of the DEM (setup_grass_environment) (Fig 1, S2 File). Second, the DEM and site locations are read

**Table 1. Input data for openSTARS.**

| Data | Mandatory or optional | Format | Description |
|------|----------------------|--------|-------------|
| digital elevation model (DEM) | mandatory | raster | elevation data needed to derive the stream network and delineate catchment boundaries |
| sampling sites | mandatory | vector | locations of the sampling sites |
| streams | optional | vector | stream network to be burnt into the DEM to guide the derived one |
| prediction sites | optional | vector | locations of sites where model predictions will be generated |
| potential predictors | optional | raster or vector | spatial datasets used to calculate predictor variables for use in the SSN model |
| measurements | optional | table (e.g. txt, csv) | measurements at the sampling sites (dependent variables and optionally potential predictors) |

into the GRASS location, along with optional data including a stream network in vector format, maps of potential predictors and prediction sites using import_data (Fig 1A; Table 1).

**Derive and clean stream network.** The preprocessing process starts by deriving a stream network from the DEM (derive_streams; Fig 1B). In this step, the optional stream network can be burnt into the DEM by a given number of meters guiding the DEM-derived streams to this network. The spatial resolution of the network (i.e. how fine the branching of the network is) can be tuned using the parameters accum_threshold and min_stream_length, which represent the minimum number of accumulated raster cells for delineating a stream line and the minimum stream length in DEM raster cells, respectively. If the resolution of the derived network is too fine or coarse this step can be repeated, or other tools can be used to determine an optimal threshold for a given stream network [21]. Next, the streams should be checked for complex confluences (check_complex_confluences) and corrected if necessary (correct_complex_confluences). The latter function moves the downstream node of one stream segment a fraction of the DEM cell size upstream, creating a tiny artificial segment in between the new and old nodes (Fig 1B). As an optional feature, artificial stream segments that flow through lakes and reservoirs can be deleted to create separate unconnected stream networks (delete_lakes). Once the streams have been topologically corrected, several attributes necessary for SSN modelling including reach contributing area (RCA, i.e. the land area adjacent to each segment that provides lateral overland flow) and catchment areas must be calculated using calc_edges [14] resulting in a new vector map of streams called 'edges'.

**Prepare sampling and prediction sites.** The site locations are cleaned (i.e. snapped to the edges if they do not exactly intersect line segments (calc_sites; Fig 1C). Additionally, attributes necessary for SSN modelling are assigned and a new vector map 'sites' is created. The first step is necessary because of frequent mismatches between site and stream locations due to GPS imprecision, the need to represent three-dimensional streams as lines in a GIS, or when deriving streams from a raster-based DEM. The column 'dist' in the sites' attribute table gives the distance a point was moved in map units. A maximum distance can be provided as an argument in calc_sites, and sites exceeding this distance will be deleted. If a large fraction of sites is moved long distances, this may indicate a too coarse spatial resolution of the stream network.

The calc_prediction_sites function (Fig 1C) allows to automatically create prediction sites for use in SSN modelling. The user specifies the number of prediction sites to be created or the distance between sites. Prediction sites are created evenly along all or selected networks in the data set with identical distances from downstream to upstream sites.

**Derive potential predictors.** Predictor variables are commonly used in SSN models to represent characteristics thought to influence the response (e.g. water quality or organism abundance). These must be assigned to the sampling and prediction sites attribute tables (Fig 1D). For approximate assignment as in the STARS toolbox, calc_attributes_edges summarises

values within the RCA and the catchment of the downstream node of each edge. Then values are assigned to the sites based on their position on the line segment using calculate_attributes_sites_approx. The second option is to derive exact catchments for each site and then summarise predictor values within the catchments (calc_attributes_sites_exact). This can be computationally intensive and take considerably longer than the approximation when there is a large number of sites.

**Export data.** The optional merge_sites_with_measurements function is used to reduce the computational resources needed to process repeated measurements at a single location (Fig 1E). Before the data are exported, a table of measurement data containing repeated measurements can be merged to the sites attribute table. A new vector point feature is generated for each repeated measurement, which contains the static predictor variables and other attributes generated in the preprocessing steps (Fig 1B–1D). Note that time-varying predictor variables will need to be generated after this step. Finally, export_ssn saves the processed data to a new local directory (a '.ssn object'), which contains streams and sample sites as shape files ('edges.shp', 'sites.shp', respectively) and optionally prediction sites, as well as topological relationships stored in text files ('netX.dat'), with the naming conventions and formats required by SSNn (Fig 1E).

### Application example and comparison with STARS toolbox

We compare the openSTARS and STARS (in ArcGIS version 10.6) output for an analysis based on 39 monitoring sites in Southern Germany (Baden-Württemberg). Point coordinates were provided by the State Environment Agency Baden Württemberg (LUBW), the DEM was provided by the European Environment Agency [22] and a stream network by the German Federal Institute of Hydrology (www.wasserblick.net). As examples for predictor maps we used the share of arable land use (vector format) in the sites' catchments (based on ATKIS land cover data [23]).

STARS requires a stream network in vector format and so we used (i) the one burnt into the network in openSTARS and (ii) the one derived from the DEM by openSTARS that exhibited a higher resolution. The results of the two tools were inspected visually and by systematically comparing the calculated catchment sizes of the sampling sites and the area of arable land use.

### Results

openSTARS and STARS yielded very similar results with regard to the position of sites snapped to the edges (Fig 2). The degree of small tributaries of the derived steam network of openSTARS depends on the choice of the parameters threshold in derive_streams and was adjusted to minimize the snapping distance of sites to an edge. However, the stream courses of both tools match.

The derived attributes catchment size and area of arable land use within the catchments of the sites derived with the two tools are very similar (correlation coefficients between the attributes calculated with STARS and openSTARS for the sites: 0.97 and 0.98, respectively; Fig 3), when based on the original stream network. The results were also similar when based on the derived stream network. There are only two exceptions: one site was snapped to a smaller tributary created in openSTARS, which is lacking in the streams dataset used in STARS, and in the other case the network is smaller (S1 File).

### Discussion

Despite the differences in the procedures of openSTARS and STARS, the derived catchment characteristics for sampling sites were very similar. The major conceptual difference between

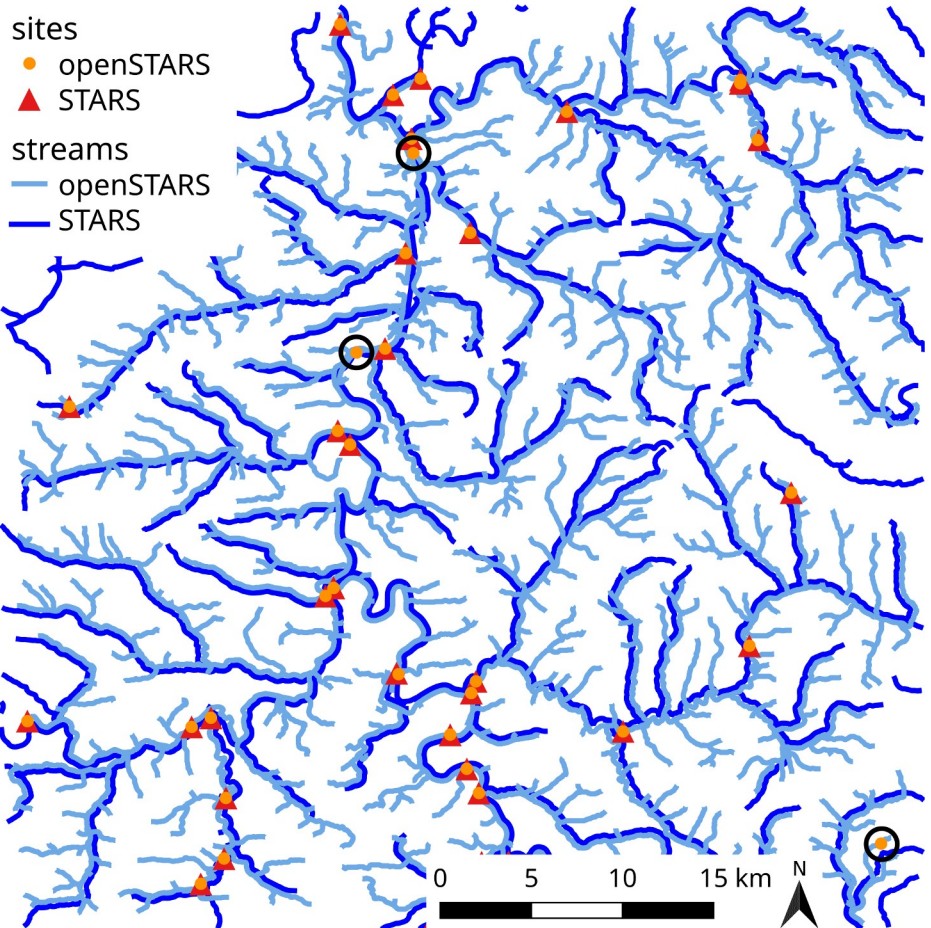

**Fig 2. Comparison of openSTARS and STARS edges and snapped sampling sites.** STARS edges with slight offset for readability. The sites marked with a dark circle were removed in STARS because their snapping distance exceeded 150 m.

the two is that the openSTARS derives the stream network from a DEM, whereas STARS relies on an existing stream network in vector format. Hence, openSTARS fills an important gap given that stream networks are either not readily available or are too coarse in many regions of the world, making them unsuitable for use in spatial statistical stream network models. More-over, existing networks in vector format often contain many topological errors that can be time consuming and difficult to correct. On the other hand, in some regions stream datasets have been topologically corrected for use in SSN modelling [24] or have been attributed with information that can be used as predictor variables [25]. Preserving such information in open-STARS would be challenging as it derives the stream network from the DEM as a new map.

Another technical difference between the tools is that calculating RCAs in STARS is based on the D8 flow direction algorithm, while openSTARS applies the more current multiple flow direction (MFD) algorithm. This may lead to differences in the calculation of RCAs and thereby in catchment areas and other potential predictors.

The requirement of non-braided streams for SSN modelling leads to another issue. In heavily modified areas (e.g. artificial drainage ditches or channels) it can be challenging to choose just one"true" stream segment. Likewise, independent of the tools used, deriving RCAs

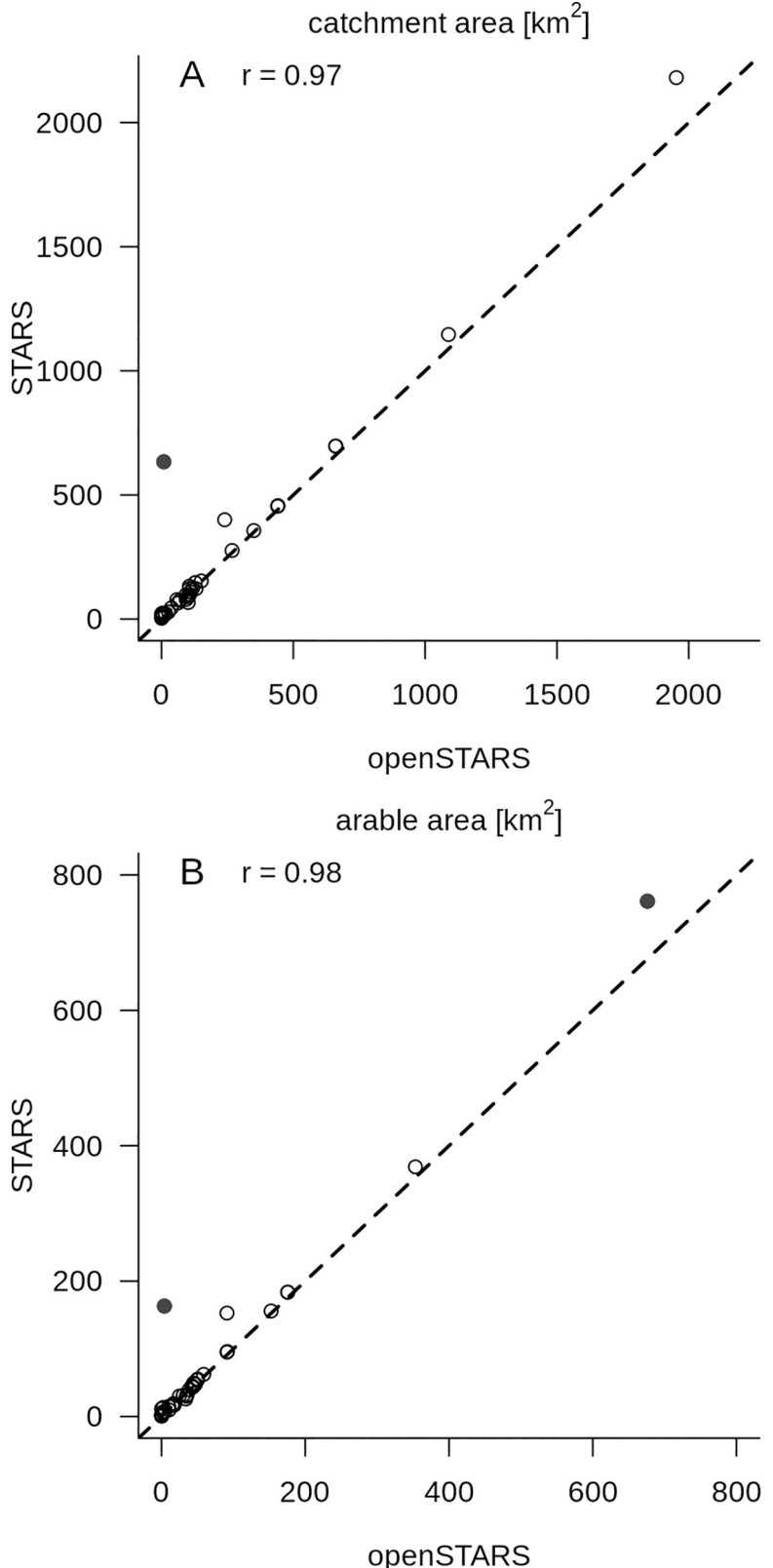

**Fig 3. Comparison of openSTARS and STARS calculated catchment attributes for the sampling sites.** A: catchment area in km$^2$; B: area of arable land use in km$^2$. r is the Pearson's correlation coefficient (including the marked outliers), the dotted line shows the 1:1 relationship, the solid black dots mark two outliers (S1 File).

and catchments in such situations may be difficult. In openSTARS, deriving the streams from the DEM ensures the absence of braided sections, and complex confluences can be corrected automatically. In STARS, the cleaning of the network is done manually or semi-automatically using the ArcGIS Topology tools, which can be very laborious depending on the size of the network and the number of such topological errors. A future openSTARS version may incorporate the option for additional manual checks and corrections.

The GRASS GIS algorithms used to derive the stream network in openSTARS (r.watershed and r.stream.extract) can fail in very flat areas where the relief energy is very low although it is deemed to outperform other algorithms [26]. In such cases, a sufficiently large DEM may provide a gradient, even if the sampling sites cover a smaller area. Additionally, burning in an existing stream network can fix this issue. However, the same issue arises for catchment delineation in the STARS toolbox.

A great advantage of openSTARS is that it relies on free and open-source GRASS GIS and R functions, unlike the original STARS toolbox for the proprietary ArcGIS software. Moreover, compared to data preparation in ArcGIS and statistical analysis in R, openSTARS unifies the complete workflow in R. Thereby it also facilitates the reproducibility and tracking of the data processing routine. In addition, a deeper understanding of GRASS or other GIS is not required.

openSTARS supports the wider application of spatial statistical modelling on stream networks, a technique that is growing in popularity for the analysis of stream data e.g. from biological or chemical monitoring. Such approaches will be particularly useful in the future, as the volume and density of data from low-cost in situ sensors continues to increase [27], and the analyses of these rich datasets may lead to new insights about stream ecosystems.

## Supporting information

**S1 File. Differences in catchment areas for STARS and openSTARS in the case study.** (PDF)

**S2 File. Complete openSTARS Workflow (commented R code).** (PDF)

## Acknowledgments

The authors thank Alan Pearse for processing the case study data in STARS and Grace Heron for testing the usability of openSTARS on various data sets.

## Author Contributions

**Conceptualization:** Mira Kattwinkel, Ralf B. Schäfer.

**Formal analysis:** Mira Kattwinkel.

**Methodology:** Mira Kattwinkel.

**Software:** Mira Kattwinkel, Eduard Szöcs.

**Validation:** Mira Kattwinkel, Erin Peterson.

**Writing – original draft:** Mira Kattwinkel.

**Writing – review & editing:** Mira Kattwinkel, Eduard Szöcs, Erin Peterson, Ralf B. Schäfer.

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
