## [Decision Letter · Decision Letter 0]

13 Jul 2020

PONE-D-20-15404

Preparing GIS data for analysis of stream monitoring data: The R package openSTARS

PLOS ONE

Dear Mira Kattwinkel

Thank you for submitting your manuscript to PLOS ONE. After careful consideration, we feel that it has merit but needs some revisions. Therefore, we invite you to submit a revised version of the manuscript that addresses the points raised during the review process.

We look forward to receiving your revised manuscript.

Kind regards,

Stoyan Nedkov

Academic Editor

PLOS ONE

Additional Editor Comments:

Dear Authors,

Dear authors, the reviewers finalized their evaluation suggesting revisions of the paper. I agree with their evaluation and suggest major revision and recommend to pay special attention on their critics. Please thoroughly address all reviewer comments in your reply and in the revised version of your manuscript.

Journal Requirements:

"MK was partly funded by the EU-INTERREG V Upper Rhine via project 1.6 SERIOR (Security-Risk-Orientation).

All other authors received no specific funding for this work."

We note that one or more of the authors are employed by a commercial company: BASF SE, Biostatistics & Data Sciences.

3.1. Please provide an amended Funding Statement declaring this commercial affiliation, as well as a statement regarding the Role of Funders in your study. If the funding organization did not play a role in the study design, data collection and analysis, decision to publish, or preparation of the manuscript and only provided financial support in the form of authors' salaries and/or research materials, please review your statements relating to the author contributions, and ensure you have specifically and accurately indicated the role(s) that these authors had in your study. You can update author roles in the Author Contributions section of the online submission form.

3.2. Please also provide an updated Competing Interests Statement declaring this commercial affiliation along with any other relevant declarations relating to employment, consultancy, patents, products in development, or marketed products, etc. 

4. We note that Figures 1, 2, S1, S2 in your submission contain map images which may be copyrighted. All PLOS content is published under the Creative Commons Attribution License (CC BY 4.0), which means that the manuscript, images, and Supporting Information files will be freely available online, and any third party is permitted to access, download, copy, distribute, and use these materials in any way, even commercially, with proper attribution. For these reasons, we cannot publish previously copyrighted maps or satellite images created using proprietary data, such as Google software (Google Maps, Street View, and Earth). For more information, see our copyright guidelines: http://journals.plos.org/plosone/s/licenses-and-copyright.

4.1.    You may seek permission from the original copyright holder of Figures 1, 2, S1, S2 to publish the content specifically under the CC BY 4.0 license.

4.2.    If you are unable to obtain permission from the original copyright holder to publish these figures under the CC BY 4.0 license or if the copyright holder’s requirements are incompatible with the CC BY 4.0 license, please either i) remove the figure or ii) supply a replacement figure that complies with the CC BY 4.0 license. Please check copyright information on all replacement figures and update the figure caption with source information. If applicable, please specify in the figure caption text when a figure is similar but not identical to the original image and is therefore for illustrative purposes only.

5. Please ensure that you refer to Figure 2 and 3 in your text as, if accepted, production will need this reference to link the reader to the figure.

Reviewers' comments:

Reviewer's Responses to Questions

**Comments to the Author**

1. Is the manuscript technically sound, and do the data support the conclusions?

Reviewer #1: Yes

Reviewer #2: Yes

2. Has the statistical analysis been performed appropriately and rigorously? 

Reviewer #1: No

Reviewer #2: Yes

3. Have the authors made all data underlying the findings in their manuscript fully available?

Reviewer #1: Yes

Reviewer #2: Yes

4. Is the manuscript presented in an intelligible fashion and written in standard English?

Reviewer #1: Yes

Reviewer #2: Yes

5. Review Comments to the Author

Reviewer #1: The submitted text "Preparing GIS data for analysis of stream monitoring data: The R package openSTARS" by Mira Kattwinkel at.al. describes a complete working licenses free library named "openStars" for R statistical environment. The software is dedicated on pre-processing of stream monitoring data for further statistical analysis, mainly by SSN. The tool is already available online and free for use. There is a manual which is easy to understand and reproduce. However, I didn't succeeded to reproduce provided example in Supporting_Information_S2. There are computational error with derive_streams(accum_threshold = 100, condition = T, clean = TRUE, burn = 10) when accum_threshold is equal to 100, failing to execute correctly calc_edges() and calc_sites(), mainly producing missing values for locID, pid and netID. The error does not exist for accum_threshold bigger than 170, but the results are different than those in supporting information. The used version of GRASS is 7.6 on R version 3.6.3 for Fedora 31.

2.The text have to be revisited because the existence of non-clear statements, such as at p.8, line 170 where the following text exists "openSTARS and STARS yielded very similar results (2)", where reference to some index (2) is confusing. Another example is the sentence on p. 9, line 182, starting with "The major conceptional difference..." where it is claimed "that the STARS derives the stream network from a DEM, whereas openSTARS relies on an existing stream network in vector format", which is opposite to the previous text.

3.According the statistical part in Paragraph 3, I think that is too brief and incomplete. Secondly, it is not clear a correlation of what and how is computed. I only could guess that outliers are removed from computation, because it is very suspicious to have r bigger that 0.98 with so large outliers. Moreover, the correlation is not suitable because samples in poor agreement may have high correlation. Likewise, the study of possible bias is also missing. As an option for revision the authors could consider any parametric or non-parametric test or at least a graphical tools, such as widely used in medicine Bland-Altman plots.

Reviewer #2: The submission itself is more or less ready to go, and explains workflow variants, and in the supplementary materials also analysis in the SSD package. The points that I feel deserve attention are that GRASS is now 7.8.3, R 4.0.2, and some of the package code is showing its age. For example, check_compl_confluences() now reports: "The command: v.to.db --quiet map=streams_v option=length type=line columns=length_new produced an error (1) during execution: ERROR: Column <length_new> exists. To overwrite, use the --overwrite flag". This may have subsequent consequences. Another is in calc_edges(), where: "Error in .prepareFastSubset(isub = isub, x = x, enclos = parent.frame(), : RHS of == is length 2 which is not 1 or nrow (1197). For robustness, no recycling is allowed (other than of length 1 RHS). Consider %in% instead. In addition: Warning message: In if (dt[stream == id, prev_str01, ] == 0) { : the condition has length > 1 and only the first element will be used" which I believe occurs from R 4.0.0, because the lengths of compared objects are now checked. These are minor issues, but would trip up users of the R package. I feel that the writing of CI code for the package code on Github with the latest released versions of R, this package and the packages it depends on, GRASS, and the GRASS extensions.

A possible vulnerability is that both GRASS and R (sp and sf packages) have adapted to changes in PROJ and GDAL, so that the handling of coordinate reference systems across the boundary between the two interfaces software systems needs checking, and relying on PROJ strings may not be as reliable as it has. So when checking the software and provided scripts for changes from version changes in R and GRASS, it would be very sensible to check whether any vulnerabilities are present as R moves from Proj4 to WKT2 string representations.</length_new>

6. PLOS authors have the option to publish the peer review history of their article (what does this mean?). If published, this will include your full peer review and any attached files.

Reviewer #1: No

Reviewer #2: No

---

## [Author Response · Author response to Decision Letter 0]

7 Aug 2020

Our response can be found in the Response to Reviewers file.

---

## [Decision Letter · Decision Letter 1]

2 Sep 2020

Preparing GIS data for analysis of stream monitoring data: The R package openSTARS

PONE-D-20-15404R1

Dear Dr. Kattwinkel,

We’re pleased to inform you that your manuscript has been judged scientifically suitable for publication and will be formally accepted for publication once it meets all outstanding technical requirements.

Kind regards,

Stoyan Nedkov

Academic Editor

PLOS ONE

Additional Editor Comments (optional):

Reviewers' comments:

Reviewer's Responses to Questions

**Comments to the Author**

1. If the authors have adequately addressed your comments raised in a previous round of review and you feel that this manuscript is now acceptable for publication, you may indicate that here to bypass the “Comments to the Author” section, enter your conflict of interest statement in the “Confidential to Editor” section, and submit your "Accept" recommendation.

Reviewer #1: All comments have been addressed

Reviewer #2: All comments have been addressed

2. Is the manuscript technically sound, and do the data support the conclusions?

Reviewer #1: Yes

Reviewer #2: Yes

3. Has the statistical analysis been performed appropriately and rigorously? 

Reviewer #1: Yes

Reviewer #2: N/A

4. Have the authors made all data underlying the findings in their manuscript fully available?

Reviewer #1: Yes

Reviewer #2: Yes

5. Is the manuscript presented in an intelligible fashion and written in standard English?

Reviewer #1: Yes

Reviewer #2: Yes

6. Review Comments to the Author

Reviewer #1: Finally, I successfully replicated the whole process as it is shown in Supporting Information 2.

The main reason for the previous issues was partly my failure - I was set different geographical projections in GRASS software. However, the error messages was completely useless and their revisions must be considered. In my opinion, the reliance on GRASS settings must be explained explicitly in the manual.

Note, that it is a technical remark for further developments and it does not impact my decision.

Reviewer #2: (No Response)

7. PLOS authors have the option to publish the peer review history of their article (what does this mean?). If published, this will include your full peer review and any attached files.

Reviewer #1: No

Reviewer #2: No

---

## [Editor Report · Acceptance letter]

9 Sep 2020

PONE-D-20-15404R1 

Preparing GIS data for analysis of stream monitoring data: The R package openSTARS 

Dear Dr. Kattwinkel:

I'm pleased to inform you that your manuscript has been deemed suitable for publication in PLOS ONE. Congratulations! Your manuscript is now with our production department. 

Kind regards, 

on behalf of

Dr. Stoyan Nedkov 

Academic Editor

PLOS ONE